# Study of the Influence of Technological Parameters on Generating Flat Part with Cylindrical Features in 3D Printing with Resin Cured by Optical Processing

**DOI:** 10.3390/polym12091941

**Published:** 2020-08-27

**Authors:** Aurel Tulcan, Mircea Dorin Vasilescu, Liliana Tulcan

**Affiliations:** 1Department of IMF, Politehnica University Timisoara, 300006 Timisoara, Romania; 2Department of MMUT, Politehnica University Timisoara, 300006 Timisoara, Romania

**Keywords:** 3D printing, additive manufacturing, surface deviation, parametric dimensioning, digital light processing, design of experiments, 3D metrology

## Abstract

The objective of this paper is to determine how the supporting structure in the DLP 3D printing process has influences on the characteristics of the flat and cylindrical surfaces. The part is printed by using the Light Control Digital (LCD) 3D printer technology. A Coordinate Measuring Machine (CMM) with contact probes is used for measuring the physical characteristics of the printed part. Two types of experiment were chosen by the authors to be made. The first part takes into consideration the influence of the density of the generated supports, at the bottom of the printed body on the characteristics of the flat surface. In parallel, it is studying the impact of support density on the dimension and quality of the surface. In the second part of the experiment, the influence of the printed supports dimension on the flatness, straightness and roundness of the printed elements were examined. It can be observed that both the numerical and dimensional optimum zones of the support structure for a prismatic element could be determined, according to two experiments carried out and the processing of the resulting data. Based on standardized data of flatness, straightness and roundness, it is possible to put in accord the values determined by measurement within the limits of standardized values.

## 1. Introduction

In both research and production, the quality obtained by the 3D printing process has an essential role in making parts or assemblies with the functional role [1,2,3,4]. At the same time, the use of this technology allows us to reduce manufacturing costs [5,6], as well as the level of pollution [7,8,9,10]. 3D printing is a relatively new technological process [11,12] that permits the generation of parts faster than other similar methods of fabrication. In the specialized literature, there are several studies related to 3D printing, among them are those dealing with the medium’s evolution in time [13,14].

The flat and profiled surfaces have an essential role, both in terms of kinematic and functional movement. The use of such an approach to generating specific features of the parts at low expense allows a reduction of both costs in design and production [15,16]. It is possible to take into consideration the analysis of the flat or round surfaces, which ensure the generation of the facets on which the movement can be achieved linearly or by rotation, with high speed and precision.

Generation of round or flat elements created by the conventional processing of injection materials or by the Fused Depositing Modeling (FDM) printing process [17,18,19], used the melted plastic as an extruded component and deposited it layer-by-layer in predetermined locations. 

In the digital light processing (DLP) [20] the optical polymerization of materials is taken into consideration for generating parts. 

An educational version of Fusion 360 is used to design the elements used in the printing process. The program is a free 3D computer-aided design (CAD) and computer-aided manufacturing (CAM) software [21]. This software allows the generation of solid body elements with a higher or lower accuracy of surfaces or shapes.

In the current technological context, in which the production of components with low manufacturing costs and the shortest manufacturing process for the production of subassemblies or components is significant. Since 3D printing ensures these desiderates, determining the dimensional accuracy of the part, as well as the form and position errors, is also essential. 3-Dimensional metrology enables the accuracy and reproducibility of the measurements in mechanical engineering [22,23,24] for measuring characteristics such as length, angles and other geometric relationships of the 3D printing part. The advantage of integrating 3D printing with 3D metrology allows us to investigate both the printing process and determine the technological modalities for achieving the previously proposed and mentioned goal.

Based on an analysis of the specialized literature, it is possible to observe that the topics proposed by the authors are new in the technical domain of 3D printing which uses the DLP technology. The same observation is for the surface quality of the 3D printed elements’ that are measured with a Coordinate Measuring Machine (CMM) with contact probes, which has higher accuracy compared to the scanning systems.

## 2. Materials and Methods

### 2.1. Considerations Regarding the Material Used for Printing and Printer Used for Making the Experimental Parts

The resin used for printing is specific to the DLP printing process, with a solidify wavelength of 405 nm [25,26]. The liquid density of resin is 1.100 × 10^3^ kg/m^3^ and the solid density of resin is 1.184 × 10^3^ kg/m^3^. In 3D printing with resin, a layer thickness of 0.05 mm is highly the most recommended for a generation. The layer exposure time recommended for the resin is 3 to 15 s for 0.02 to 0.10 mm layer thickness and the bottom exposure time is 20 to 80 s for 4 to 8 layers in the bottom position. From our experience of 3D printing with this type of resin, 50 s for the bottom eight layers and 6 s for the 0.05 mm of the layer is the exposure time recommended for use.

The printer used for the generation of the element is of the low-cost commercial model type. For this printing process, an ANYCUBIC PHOTON printer (Anycubic, Shenzhen, China) was used [27]. This type of printer uses in the printing process light emitted by using a transparent screen with a multiple point size diameter with 47 micron value. To generate the photo polymerization, an ultraviolet light source with 25-watt power and wavelength of 405 nm is used. The resolution along *Z*-axis is 1.25 µm. Enclosure with thermal propriety and automatic measurement of the temperature and humidity is proposed by the authors to be used in the printing process. This enclosure with ARDUINO automatization is made in our laboratory. These two physical elements were measured inside the printer chamber and in the enclosure. The use of an enclosure can maintain the printing temperature of the resin between 25 and 30 degrees, which is vital for a good printing process with resin.

### 2.2. Considerations Regarding the Generation of the Model Used for Making the Experimental Parts

In research, as well as in production, the quality of surfaces, volumes and precision obtained by 3D printing, has an essential role in making parts or assemblies with a functional role. 3D printing is a relatively new technological process that enables the generation of parts faster than other similar mechanical or chemical methods. The flat and profiled surfaces have an essential role, both in terms of kinematic and functional movement. The use of such an approach to generating specific features of the parts at low costs allows reductions of both design and production costs.

The dimensional and surface characteristic were analysed. This study would allow us to design a part that would include both types of features mentioned above. At this point, we have chosen to develop a complete and correct examination, for the experimental point of view. Figure 1 shows the part of the parallelepiped shape with a length of 55 mm, a width of 35 mm and a thickness of 5 mm. The part has three cylindrical holes with a diameter of 5 mm, 10 mm and 15 mm and three cylindrical bosses with a diameter of 5 mm, 10 mm and 15 mm with a height of 10 mm, all generated by specific CAD operations.

An essential step of generating the body element is the transformation of CAD mechanical structure created into a body component with a precision lower with 752 triangles or higher with 2342 triangles. Figure 2 shows the mesh generated with different refinement levels. On the left picture in Figure 2a has presented the body with the lowest refinement level of the mesh at 752 triangles and on the right view, Figure 2b the body component with the highest refinement level of the mesh at 2342 triangles.

It is possible to see that the resolution can be changed more or less, for the surfaces that have a circular section, when generating the body element structure.

An essential aspect, which makes the 3D measurements possible, was the location of the cylindrical features (holes and bosses) on the part surface. The minimum distances between their cylindrical surfaces are 3 mm, taking into account the 2 mm in diameter of the probe stylus tip diameter installed on the probe head of the CMM used to measure the part.

### 2.3. Considerations Regarding the Measuring of the 3D Printed Probe

For the measurement of physical characteristics such as length, angles and other geometric elements in relation to the 3D printing part, a DEA Global Advantage 7.10.7 CMM with contact probes was used [28]. This CMM had a calibration certificate with Maximum Permissible Error (MPE_E_) for length measurement: MPE_E_ = 1.9 + L/300 μm [28,29]. The software metrology used to create and execute measurement routines was PC-DMIS 2019 R2 [30]. The first step in creating the measurement routine consisted in the mathematical alignment of the physical part, installed on the CMM, with the CAD model imported into PC-DMIS software and setting the origin according to the CAD model, shown in Figure 3a. The printed part was clamped in a parallel vise with three pins [31], shown in Figure 3b. To be able to fix the part between the three pins of the vise, the edge of the solid bottom layers of the printed part was removed by cutting. Figure 3c shows the definition of the part surfaces and the measuring planes, in which the cylindrical features have been assessed. The definition of the part probing strategy is presented in Appendix A. The measured feature (e.g., plane, line and circle), is defined for each of the physical surfaces, Z-coordinate of the measuring plane of the feature and the number of the probing (contact) points. The probing point patterns obtained from measuring for the flat and cylindrical surfaces and the deviations in each probing point are presented in Figure 3d. The arrows indicate the direction of the deviations and it is possible to observe how the dimension could produce a modification in the structure of the 3D printed part.

### 2.4. Considerations Regarding the Method Used in the Experiment

The method used in the experiment is based on successive printed and measuring tests, set to minimize the number of 3D printed parts, as well as the number of measurements. 

The first stage deals with the design of the body element. At this stage, based on the previous author’s experience, the behaviour of flat surfaces generated was first considered depending on how they are produced. Subsequently, after determining an optimal area, the next step was to identify the optimal support settings, concerning the shape of the supports of the body element subjected to the printing process, based on a factorial experiment. 

The second stage consisted of the measurement program’s creation using the 3D model for measuring the printed parts. Following the printed parts’ generation in the first stage, corrections of the measurement program and verification of the way of carrying out the measurement process were made. 

The first complete experimental program was carried out, which involved establishing the number of supports that were damaged or broken as a result of the printing process. It was also identified how the breaking of some supports, influences the dimension in the measurement process. The dimension is studied both in terms of sizes and in terms of geometric position. 

The second completed experiment takes into consideration how the geometric dimension and the modifications in the density of the supports could influence the measured elements of the printed part. For this consideration, a factorial analysis of a specific experimental research process is taken into account by the authors. Finally, this data was put in concordance with the standard data from ISO 2768-2 General Tolerances.

## 3. Results

From an experimental point of view, the authors split the experiment into two parts, due to the volume of data, which must be put in the experiment and information from the specialized literature and our experience. 

### 3.1. Results on the Supports Damaged or Broken During the DLP 3D Printing Process

For a reasonable solution in the first printing experiment, the authors take into consideration the automatic generation, using the specific value presented in Appendix A.

The method of counting the total number of supports used to determine the number of these elements in the generation plane is the paper method. This is based on a paper strip with 0.1 mm thickness used to determine if there is a gap (separation point) between the supports and the part. The picture of the printed structure, for the heavy density solution with 30, 40 and 50% support density, photographed from the side, are presented in Figure 4.

It is essential to observe that in the first and second columns of Appendix A, there is a direct correlation between values. The same situation is for columns three and four. This observation of the experiment imposes to select two of these and we decided to choose the diameter of the contact surface and the supports density (columns two and three). With this observation, in the program Statgraphics Centurion [32] has created a response surface design (3^2^) [33]. The authors have selected two experimental (independent) factors and two responses (dependent) variables for this part of the experiment. The independent factors are the diameter of the contact surface of the supports and support density. As dependent variables, there are the number of broken supports and the flatness of the surface A shown in Figure 3c. For each independent factor, the three level’s minimum, medium and maximum were chosen.

The experiment started with a 30% support density. In Appendix A, all supports were broken (100% percentage) for the diameter of the contact surface of 0.8 mm. For the other two values of 1.2 mm and 1.6 mm of the diameter of the contact surface, the percentage of the broken supports is 36.95% and 26.78%. Due to the large number of the broken supports at 30% support density, for a 3^2^ factorial experiment, 40%, 50% and 60% support density values were chosen.

The first analysis of the factorial experiment starts with the broken supports dependent variable. In Table 1 presents the analysis of variance [32,34] for broken supports. If the *p*-value for independent factors is less than 0.05, for a 95% confidence level, the independent factors significantly affect the outcome or response of the dependent variable. It is possible to see that the number of broken supports is significantly affected by the diameter of the contact surface of them follows by the support density and the second order effect that are square diameter contact and interaction between diameter contact and support density. The influence of the factors that affect the number of broken supports is visible in the standardized Pareto Chart, presented in Appendix A. All the bars exceeding the vertical line show that the factors significantly affect the broken supports as a dependent variable.

The regression of the fitted model for the broken supports is presented in Equation (1):**Broken supports** = 1150.92 − 1116.04 × Dia_contact − 14.125 × Support density + 275.0 × (Dia_contact)^2^ + 6.6875 × Dia_contact × Support density + 0.04 × (Support density)^2^.(1)

Increasing the diameter of the contact supports and support density leads to a reduction in the number of broken supports. Appendix A shows the estimated response surface and Figure 5, the contour of a determined response surface for the number of broken supports. The optimum value is for 51% support density and 1.43 mm contact diameter. The estimated response surface is important to observe the position of the minimum or maximum point of the surface. The value is determinate by the measurement of the point in the Figure 5 in which is presented the contour of the determinate response surface.

The second part of this analysis of the factorial experiment continues with the levelness of the flat surface of the part. This surface is noted in Figure 3c with A. The analysis of variance for flatness is presented in Table 2. The flatness of the surface A is influenced, for a 95% confidence level, only by the diameter of the contact surface of the supports that affect significantly this experiment. The Pareto Chart for flatness is possible to be observed in Appendix A. This fact is indicated in the table by the *p*-value that is less than 0.05 only for the diameter of the contact surface.

Increasing the diameter of the contact supports leads to a reduction of the surface flatness. For the support density, the minimum of flatness with a value of 0.49 mm is achieved at a value of support density of 48.5%. Appendix A shown the estimated response surface and Figure 6, the contour of the estimated response surface for the flatness. The estimated response surface is important to observe the position of the minimum or maximum point of the surface. The value is determinate by the measurement of the point in the Figure 6 in which is presented the contour of the determinate response surface.

The regression of the fitted model for the flatness is presented in Equation (2):**Flatness** = 4.19647 − 1.44271 × Dia_contact − 0.105042 × Support density + 0.354167 × (Dia_contact)^2^ + 0.0061875 × Dia_contact × Support density + 0.00100167 × (Support density)^2^.(2)

From the estimated response surface for broken supports, there is a minimum value of broken supports at 50% supports density and 1.4 mm diameter of the contact.

For the flatness at the automatic generation of supports, a minimum value is possible to obtain at 48% support density and 1.6 mm contact diameter.

It is important to observe that for 1.2 mm contact diameter for supports in the automatic generation system the contact depth is 0.2 mm and for 1.6 mm in the same situation the value is 0.3 mm. From mathematical interpolation of the contact diameter value between 1.2 mm and 1.6 mm, results in a value of 0.25 mm for the contact depth of 1.4 mm.

### 3.2. Results on the Supports Constructive Dimension in DLP 3D Printing Part Process

The study of the influence of dimensional geometry of the supports is made in the second part of this experiment. In this part, for each generating support structure, the automatic generating supports are corrected with a manual version, to repair the errors of the automatic supports generation. The data for the second experiment is shown in Appendix A. The factorial analysis is a Box-Behnken design, which will study the effects of three factors in 15 runs in a single block. The order of the experiments was randomized.

An important aspect considered in the second part of the experiment was that of generating a sample that was as close as possible to the final values desired to obtain at the realization of the printed part. Because the printing is made in a vertical direction, along the *Z*-axis, for this direction, there is no modification of dimensional values. Appendix A shows the dimensions that were subjected to the correction process. In Appendix A is presented the generation mode of arrangement of the sections of the quotas marker.

#### 3.2.1. Results Concerning the Flatness of the Body Surface with a Different Dimension of the Supports for DLP 3D Printing Part Process

In the experimental process carried out in this phase, a measurement plan was made. The plane surface A, shown in Figure 7, was measured in 110 contact points, according to Appendix A.

The experimental value was centralized in Appendix A and the analysis of variance for flatness was presented in Appendix A. From this, only the A, C, AA, BB, BC, and square CC factors have a major influence. The others have a minor influence which can be ignored. The *p*-value from Appendix A, less than 0.05, indicates, for a 95% confidence level [32,33,34], that only the independent factors A and C, and the interactions AA, BB, BC and CC significantly affect the response of the dependent variable. The influence factors that affect the flatness can also be observed in the standardized Pareto Chart, presented in Figure 8. The regression of the fitted model for the flatness, for the second experiment, is presented in Equation (3):
**Flatness** = 9.7485 − 0.313337 × Support density − 2.9575 × Contact depth − 1.07219 × Dia.contact + 0.00293375 × (Support density)^2^ + 0.0335 × Support density × Contact depth + 0.0034375 × Support density × Dia.contact + 12.6125 × (Contact depth)^2^ − 3.575 × Contact depth × Dia.contact + 0.596094 × (Dia.contact)^2^.(3)

In the estimated response surface from Appendix A it is important to note the position of the minimum or maximum point of the surface. The value is determined by measurement of the point in Figure 9, Figure 10 and Figure 11, in which is presented the contour of the determined response surface.

For a better presentation of flatness data resulting from the factorial experiment, they were centralized in Table 3.

For the flatness aspects, the following details have been observed:The impact of the support density and contact diameter is greater than the contact depth of the geometry of the support;Minimum flatness could be obtained at a point positioned at 51.50% density of supports, 0.23 mm contact depth mean for the three values and 1.35 mm contact diameter, with a flatness value around 0.40 mm;The minimum value of the flatness, obtained for the fifteen runs of the experiment are indicated in Appendix A, is 0.495 mm, for the levels of 50% support density, 0.3 mm contact depth and 1.6 mm contact diameter.

#### 3.2.2. Results Concerning the Straightness of the Body Surface with a Different Dimension of the Supports for DLP 3D Printing Part Process

For the experimental process carried out in this phase, regarding the straightness of four contours of the surface body, a measurement plan was made. The measurement of the contours was made along the X-axis and *Y*-axis. These contours noted with X1, X2 and Y1, Y2, are shown, generally, in Figure 3c, and more detailed in Figure 12. The measure was made with an offset of 0.8 mm from the external edges of the surface A, shown in Figure 12. The probing point is presented in Appendix A. From the origin of the coordinate system of the part, the contour X1 was measured along *Y*-axis, with an offset of X = 0.8 mm. The contour Y1 was measured along X-axis, with an offset of Y = 0.8 mm from the origin of the part. A similar principle was used for the contours X2 and Y2.

The analysis of variance for straightness is made for all four contours. Due to the limited number of pages, the results are not presented in this paper. Each contour is displayed with the standardized Pareto Chart, which indicates the main factors and their interactions, which significantly affect the contour straightness.

The analysis of the results of this experiment starts with the interpretation of the results obtained for the contours X1 and X2. In the second part of this experiment, the analysis will continue with the interpretation of the results obtained for the contours Y1 and Y2.

##### Contour X1

The influence factors that affect the straightness can also be seen in the standardized Pareto Chart, presented in Figure 13. All the bars exceeding the vertical line show that the factors significantly affect the straightness of the dependent variable. It is possible to observe that only the diameter contact surface does not have a significant influence on the X1 contour straightness.

For a better presentation of straightness data resulting from the factorial experiment, the resulting data were centralized in Table 4.

In the estimated response surface from Appendix A, it is important to observe the position of the minimum or maximum point of the surface. The value is determined by the measurement of the point in Figure 14, Figure 15 and Figure 16, in which is presented the contour of the determined response surface.

The regression of the fitted model for the contour X1 straightness is presented in Equation (4):**Straightness_X1** = 3.27675 − 0.0993125 × Support density − 3.14 × Contact depth − 0.488438 × Dia.contact + 0.00089375 × (Support density)^2^ + 0.0275 × Support density × Contact depth + 0.0021875 × Support density × Dia.contact + 7.3125 × (Contact depth)^2^ − 1.075 × Contact depth × Dia.contact + 0.242969 × (Dia.contact)^2^.(4)

##### Contour X2

The influence factors that affect the straightness can also be seen in the standardized Pareto Chart, presented in Figure 17. It can be observed that only the second order effect of the support density and contact depth have a significant influence on the X2 contour straightness.

The regression of the fitted model for the contour X2 straightness is presented in Equation (5):**Straightness_X2** = 4.02087 − 0.114313 × Support density − 3.62625 × Contact depth − 0.975 × Dia.contact + 0.00098625 × (Support density)^2^ + 0.03225 × Support density × Contact depth + 0.006125 × Support density × Dia.contact + 9.6125 × (Contact depth)^2^ − 1.625 × Contact depth × Dia.contact + 0.399219 × (Dia.contact)^2^.(5)

For a better presentation of straightness data resulting from the factorial experiment, this information was centralized in Table 5.

In the estimated response surface from Appendix A, it is important to remark the position of the minimum or maximum point of the surface. The value is determined by measurement of the point in Figure 18, Figure 19 and Figure 20, which show the contour of the determined response surface.

The second experiment looks into studying the influence of the dimension of the supports on the straightness of the contours located near the body surface edges, here we have noted the following:There are differences between the two directions of measuring the straightness. The differences are produced by the existence of the holes in the body on one side of the parts under which there are no supports. These aspects determined that the straightness (0.1 mm) is better on the side of the part with cylinder (supported) compared to the straightness (0.13 mm) on the opposite part, which is not supported.The optimum dimension of the supports, for X1 contour, is at 51% density of structure for sustainment of the body and for supports geometry, at 0.21 mm contact depth point and 1.23 mm of contact diameter mean value.The optimum dimension of the supports, for X2 contour, is at 51% density of structure for sustainment of the body and for supports geometry, at 0.20 mm contact depth point and 1.22 mm of contact diameter.The minimum value of the straightness along the *Y*-axis is 0.120 mm, shown in Appendix A, which is obtained for X1 contour at 50% support density, 0.2 mm contact depth and 1.2 mm contact diameter.

##### Contour Y1

The influence factors that affect the straightness can also be seen in the standardized Pareto Chart, presented in Figure 21. It is possible to observe that only the square interaction of support density and contact depth, respectively the interaction between the contact depth with diameter contact has a significant influence on Y1 contour straightness.

The regression of the fitted model for the contour Y1 straightness is presented in Equation (6):**Straightness_Y1** = 9.41137 − 0.307325 × Support density − 6.25625 × Contact depth − 0.925313 × Dia.contact + 0.00271375 × (Support density)^2^ + 0.097 × Support density × Contact depth + 0.0095 × Support density × Dia.contact + 18.8125 × (Contact depth)^2^ − 5.44375 × Contact depth × Dia.contact + 0.594531 × (Dia.contact)^2^.(6)

For a better understanding of straightness data resulting from the factorial experiment, the corresponding data were centralized in the Table 6. 

In the estimated response surface from Appendix A, it is important to observe the position of the minimum or maximum point of the surface. The value is determined by measurement of the point in Figure 22, Figure 23 and Figure 24, in which is presented the contour of the determined response surface.

##### Contour Y2

The influence factors that affect the straightness can also be seen in the standardized Pareto Chart, presented in Figure 25. It can be observed that support density and the square interaction of the support density and contact diameter, respectively the interactions between the support density and diameter contact, contact depth and support density have a significant influence on Y2 contour straightness.

For a better presentation of straightness data resulting from the factorial experiment, they were centralized in Table 7.

The estimated response surface from Appendix A, it is important to observe the position of the minimum or maximum point of the surface. The value is determined by measurement of the point in Figure 26, Figure 27 and Figure 28, in which is presented the contour of the determined response surface.

The regression of the fitted model for the contour Y2 straightness is presented in Equation (7):**Straightness_Y2** = 7.22063 − 0.24275 × Support density − 4.59875 × Contact depth − 0.270312 × Dia.contact + 0.00224 × (Support density)^2^ + 0.06225 × Support density × Contact depth + 0.0013125 × Support density × Dia.contact + 12.025 × (Contact depth)^2^ − 3.075 × Contact depth × Dia.contact + 0.323437 × (Dia.contact)^2^.(7)

In the second experiment, in which studies the influence of the dimension of the supports on the straightness of the contours located near the body surface edges, it is possible to observe that:There are differences between the two directions of measuring the straightness. These differentiations are produced by the existence of the holes in the body in one side of the parts under there are no supports. This aspect determined that the straightness (0.25 mm) be better on the side of the part with a cylinder (supported) compared to the straightness (0.29 mm) on the opposite side of the part, with the hole in the surface not supported.The optimum dimension of the supports, for Y1 contour the density of the structure is at 51% density for sustainment of the body and for supports geometry, at 0.22 mm contact depth point and 1.40 mm of contact diameter mean value.The optimum dimension of the supports, for Y2 contour, is at 51% density of structure for sustained the body and for supports geometry, at 0.22 mm contact depth point and 1.40 mm of contact diameter.The minimum value of the straightness along the *X*-axis is 0.252 mm, shown in Appendix A, which is possible to result for Y1 contour at 50% support density, 0.3 mm contact depth and 1.6 mm contact diameter.

#### 3.2.3. Results Concerning the Roundness of the Cylindrical Features with a Different size of the Supports for DLP 3D Printing Part Process

For a better presentation of the experimental process carried out in this phase, a measurement plan was made. The hole H15 and cylinder Cy15, indicated in Figure 29, are taken into consideration.

For all the round features (holes and outside cylinders indicated in Figure 3c) was done the Analysis of Variance for roundness. Due to the limited number of pages, the author presented in the paper only a part of the information.

Regarding the roundness of the holes, in the following, there are presented only the results for the roundness of the hole with a 15 mm diameter, marked H15 according to Figure 29. Figure 30 comprises the standardized Pareto Chart were the factors A, B, C and square CC second order effect significantly affect the contour roundness of the hole. The others interactions have a minor influence and can be ignored.

The regression of the fitted model for the roundness of the hole with a 15 mm diameter is presented in Equation (8):**Roundness Hole_15** = 0.19625 + 0.0055875 × Suport density − 0.3125 × Contact depth − 0.398438 × Dia.contact − 0.00003875 × (Suport density)^2^ − 0.0025 × Suport density × Contact depth − 0.0000625 × Suport density × Dia.contact − 0.0625 × (Contact depth)^2^ + 0.3 × Contact depth × Dia.contact + 0.132031 × (Dia.contact)^2^.(8)

For a better presentation of roundness data resulting from the factorial experiment, this data was centralized in the Table 8.

The estimated response surface from Appendix A is important to observe the position of the minimum or maximum point of the surface. The value determined by measurement of the point in Figure 31, Figure 32 and Figure 33, in which is possible to be seen the presented of the contour of the determined response surface.

In the second experiment in which it is studying the influence of the dimension of the supports on the roundness of the cylindrical holes, it can observe that:The impact of the three main factors and the second order effect of the diameter contact is more significant than the factors interactions and others second order effects of the geometry of the support;Minimum roundness is possible to be obtained in a point positioned at 40% density of supports, 0.3 mm contact depth and 1.20 mm contact diameter, with a roundness value around 0.045 mm.

Regarding the roundness of the external cylindrical features, the following will present just the results of the roundness of the cylinder with 15 mm diameter, determinate in measuring plane two marked MP2, according to the Figure 29, with Z-coordinate defined in Appendix A. Figure 34 shows the standardized Pareto Chart, which indicates that only the interaction between the diameter contact and contact depth significantly effects of the contour roundness of the cylinder.

The regression of the fitted model for the roundness of the cylinder with 15 mm diameter is determined in the measuring plane two (MP2) and presented in Equation (9):**Roundness Circle_15_MP2** = 0.555375 − 0.0188708 × Suport density + 1.10292 × Contact depth − 0.181875 × Dia.contact + 0.000155833 × (Suport density)^2^ − 0.0165 × Suport density × Contact depth + 0.0043125 × Suport density × Dia.contact + 1.30833 × (Contact depth)^2^ − 0.70625 × Contact depth × Dia.contact + 0.0395833 × (Dia.contact)^2^.(9)

For a better understanding of roundness data resulting from the factorial experiment, this data was centralized in Table 9.

The estimated response surface from Appendix A, it is important to observe the position of the minimum or maximum point of the surface. The value determined by measurement of the point in Figure 35, Figure 36 and Figure 37, in which is possible to be seen the presented of the contour of the determined response surface.

The second part of this experiment study, which looks into the influence of the dimension of the supports on the roundness of the external cylindrical features, measured as circles in three measuring planes. The following conclusion has been observed:For the 15 mm cylinder, measured as the circle in measuring plane MP2, the impact of the interaction between the contact depth and contact diameter is more significant than the main factors impact, their interactions and their second order effects;Minimum roundness can be obtained in a point positioned at 59% density of supports, 0.28 mm contact depth and 1.2 mm contact diameter, with a roundness value around 0.015 mm;

Regarding all cylindrical features (holes and external cylinders), measured as internal or external circles, the following conclusion has been observed:The minimum value of the roundness, shown in Appendix A is 0.018 mm for the hole of 5 mm, 0.043 mm for the hole of 10 mm, at 50% density, 0.1 mm contact depth and 1.6 mm contact;The minimum value of the roundness for all the cylindrical features, obtained during the fifteen runs of the experiment, is shown in Appendix A and correspond at 60% support density, 0.2 mm depth and 1.6 mm diameter.

## 4. Discussion

The first experimental results indicate that the flatness in correlation with the broken supports is better for a support density from 48% to 50%. 

In the second experiment, which studies the influence of the dimension of the supports on the flatness of the part, it is possible to observe that the flatness value (0.4 mm) obtain after the correction of the supports, by position and number, is better compared to the flatness value (0.49 mm) obtained before. This observation is important for technological decisions. The second observation is that the supports density is between 50% and 52% for an optimal solution.

Based on this we can conclude that, according to the ISO 2768-2 standard [35] regarding the geometrical tolerances for features without individual tolerances indications, the value of the flatness that could be obtained. For the nominal lengths of the part over 30 mm up to 100 mm, could be included in the L tolerance class, with a standard value of 0.4 mm [35].

Based on the above observations, we can conclude that, according to the ISO 2768-2 standard [35], the value of the straightness that could be obtained, for the nominal lengths of the part over 30 mm up to 100 mm, could be included in the K tolerance class, with a standard value of 0.2 mm [35].

Based on the above observations, we can conclude that, according to the ISO 2768-2 [35] standard, the value of the roundness that could be obtained from the 3D DLP printing process can be included in the H tolerance class, with a standard value of 0.1 mm [35].

From the first experiment and respectively from the first part of the second experiment, it can be observed that for a flat surface, it is recommended that the support density with the subsequent correction of their arrangement be between 50% and 52%. Their geometry ensures optimal value for a contact diameter between 1.2 mm and 1.3 mm, respectively with a contact depth between 0.20 mm and 0.23 mm for a surface with good value following standards for straightness and flatness.

For the roundness characteristics, the optimal value obtained for a support density is between 40% and 60%. Their geometry ensures optimal value for a contact diameter with 1.2 mm, respectively a contact depth between 0.28 mm and 0.30 mm for a good value following the general standards for cylindrical features.

## Figures and Tables

**Figure 1 polymers-12-01941-f001:**
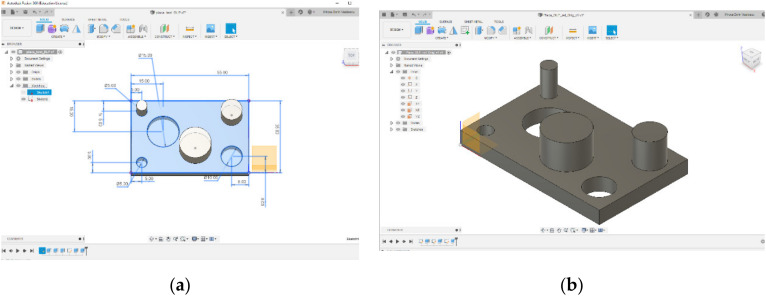
CAD part generation: (**a**) The part specifications; (**b**) The body model.

**Figure 2 polymers-12-01941-f002:**
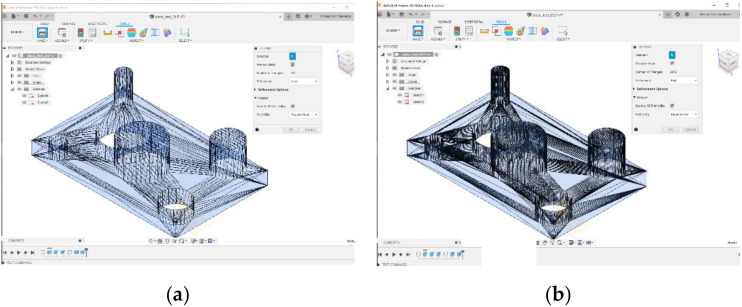
Meshing the part: (**a**) Low refinement; (**b**) High refinement.

**Figure 3 polymers-12-01941-f003:**
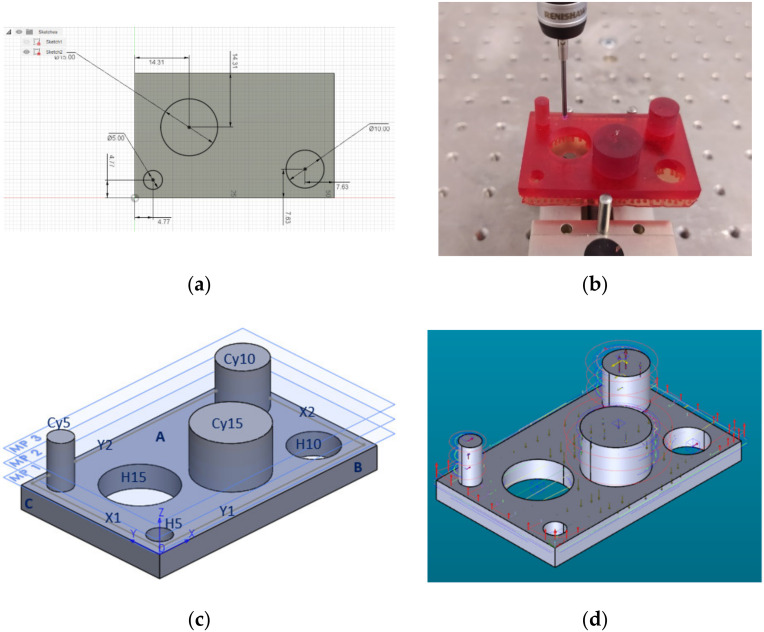
Part measuring: (**a**) 3D model origin; (**b**) Clamping the part; (**c**) Definition of the part surfaces; (**d**) Probing point’s patterns and deviations direction.

**Figure 4 polymers-12-01941-f004:**
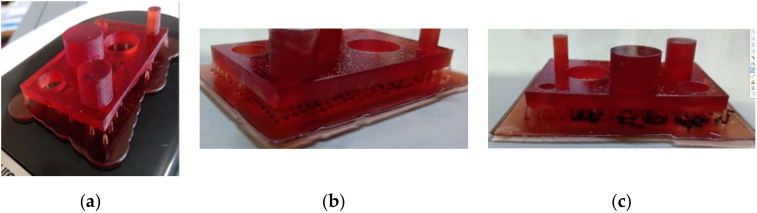
Supports type in function of heavy density: (**a**) density 30%; (**b**) density 40%; (**c**) density 50%.

**Figure 5 polymers-12-01941-f005:**
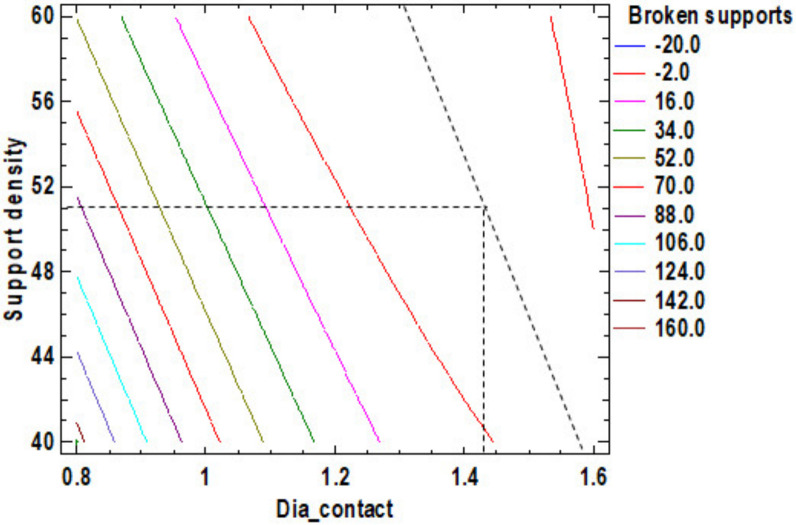
Contours of Estimated Response Surface for Broken supports.

**Figure 6 polymers-12-01941-f006:**
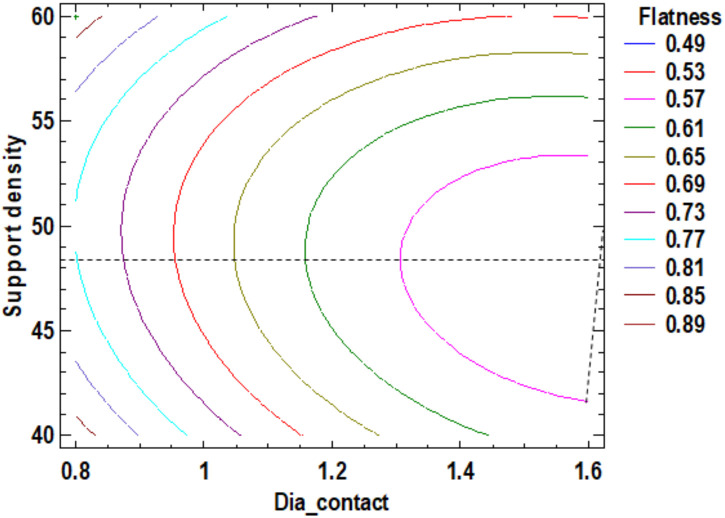
Contours of Estimated Response Surface for Flatness.

**Figure 7 polymers-12-01941-f007:**
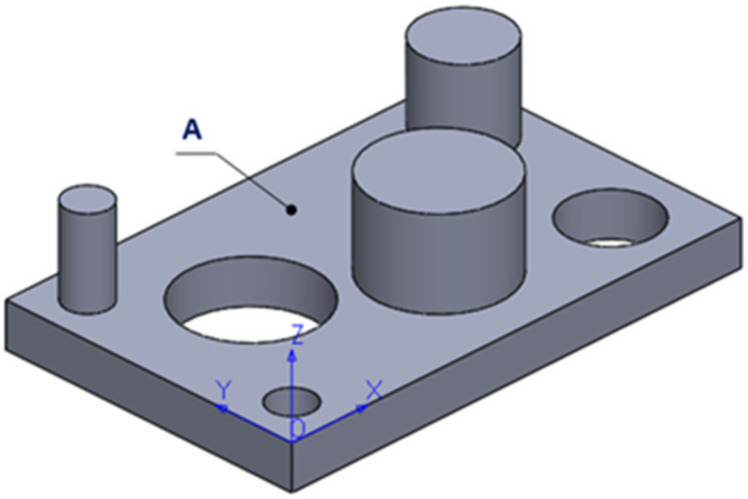
Definition of the measured plane surface.

**Figure 8 polymers-12-01941-f008:**
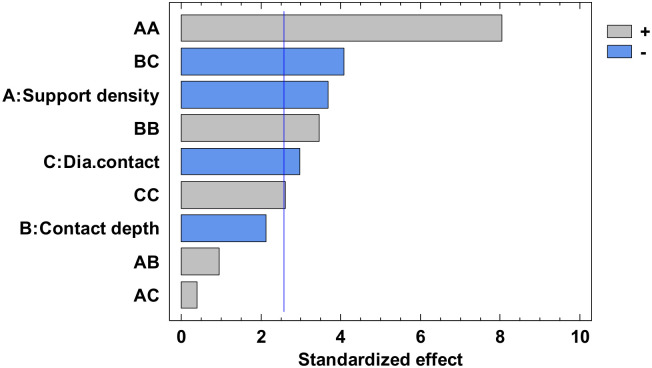
Standardized Pareto Chart for Flatness.

**Figure 9 polymers-12-01941-f009:**
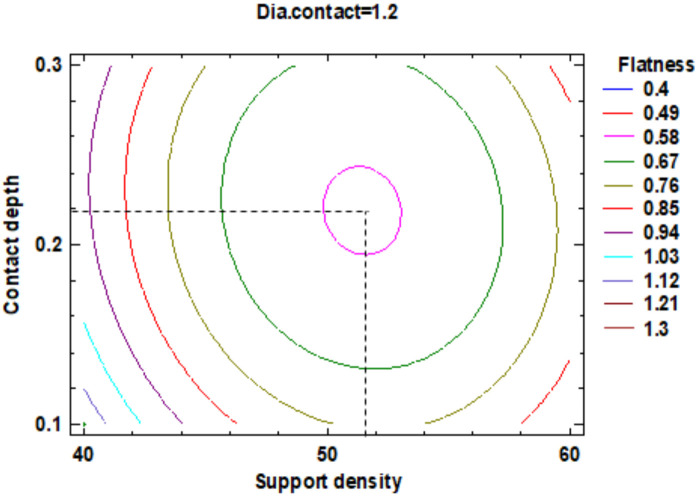
Contours of Estimated Response Surface for the Flatness and centre diameter contact.

**Figure 10 polymers-12-01941-f010:**
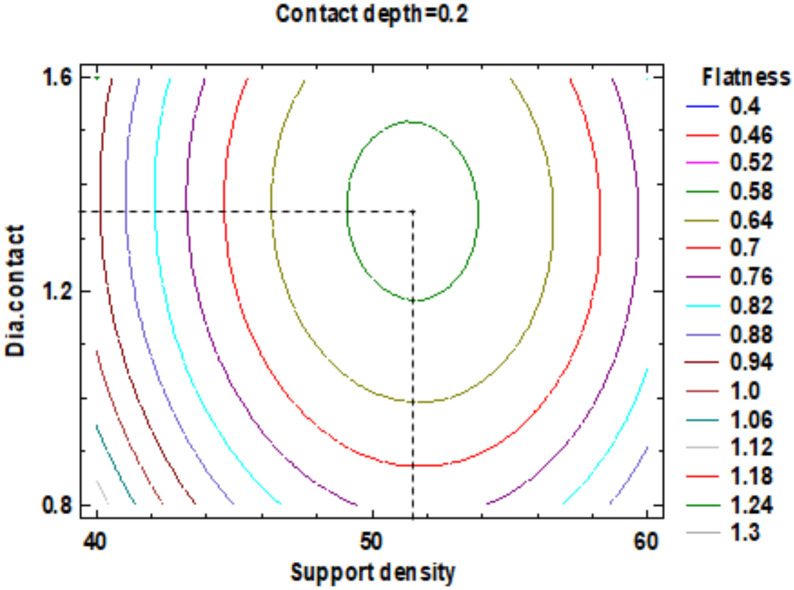
Contours of Estimated Response Surface for Flatness and centre contact depth.

**Figure 11 polymers-12-01941-f011:**
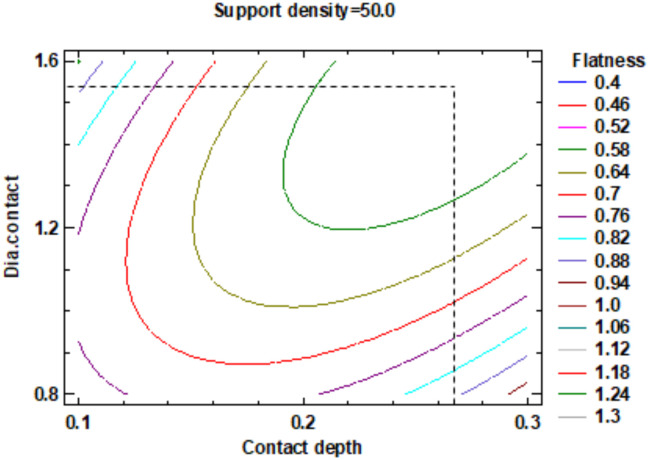
Contours of Estimated Response Surface for Flatness and centre support density.

**Figure 12 polymers-12-01941-f012:**
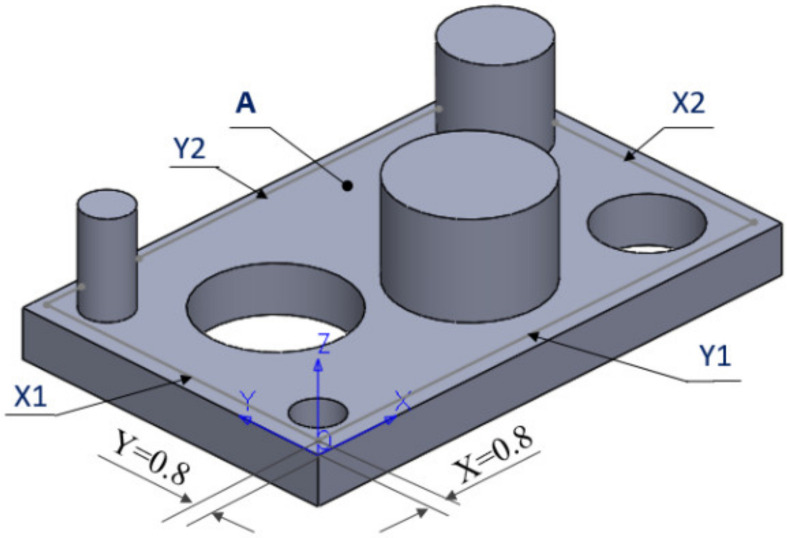
Definition of the linear contours.

**Figure 13 polymers-12-01941-f013:**
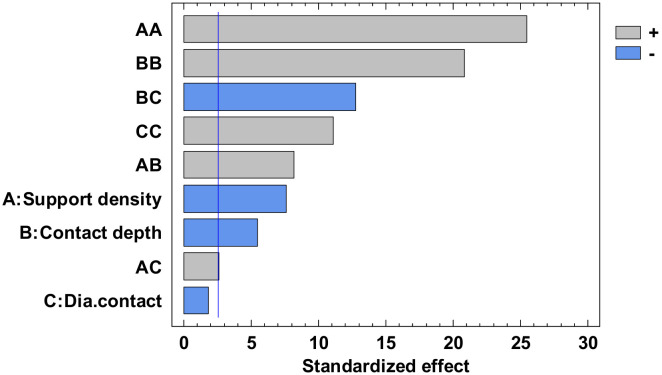
Standardized Pareto Chart for Straigtness_X1.

**Figure 14 polymers-12-01941-f014:**
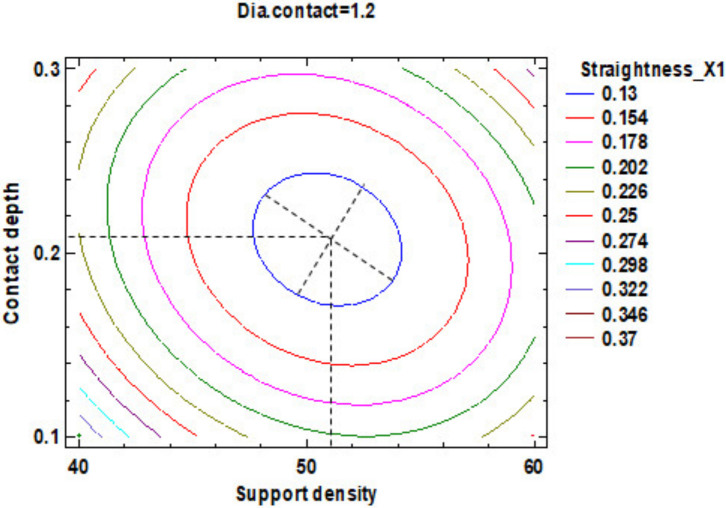
Contours of Estimated Response Surface for Straightness_X1 and centre diameter contact.

**Figure 15 polymers-12-01941-f015:**
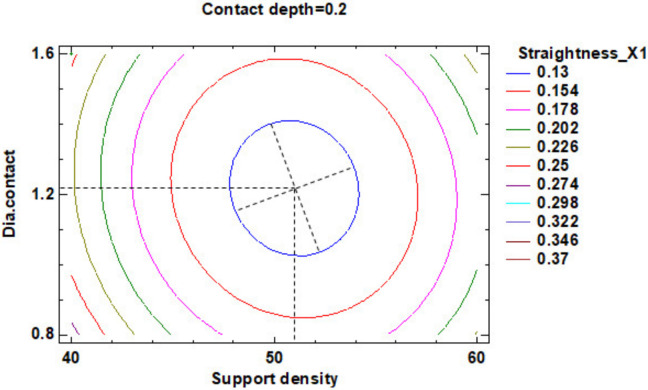
Contours of Estimated Response Surface for Straightness_X1 and centre contact depth.

**Figure 16 polymers-12-01941-f016:**
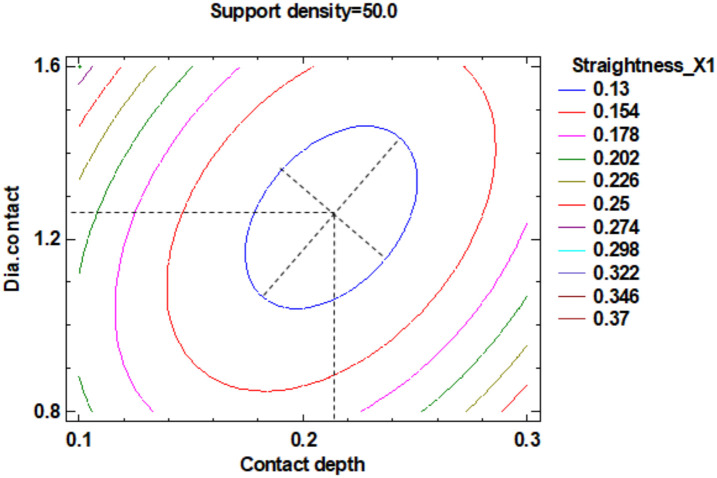
Contours of Estimated Response Surface for Straightness_X1 and centre support density.

**Figure 17 polymers-12-01941-f017:**
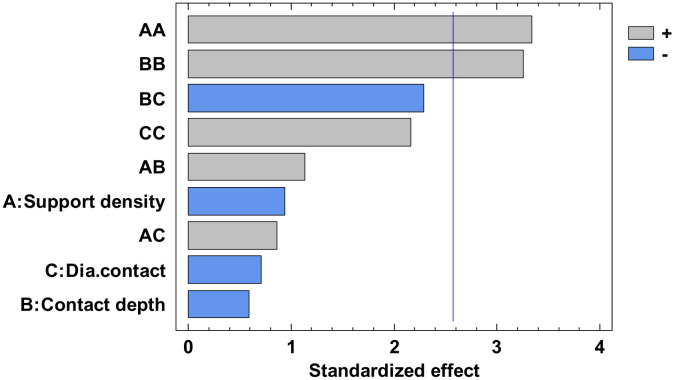
Standardized Pareto Chart for Straigtness_X2.

**Figure 18 polymers-12-01941-f018:**
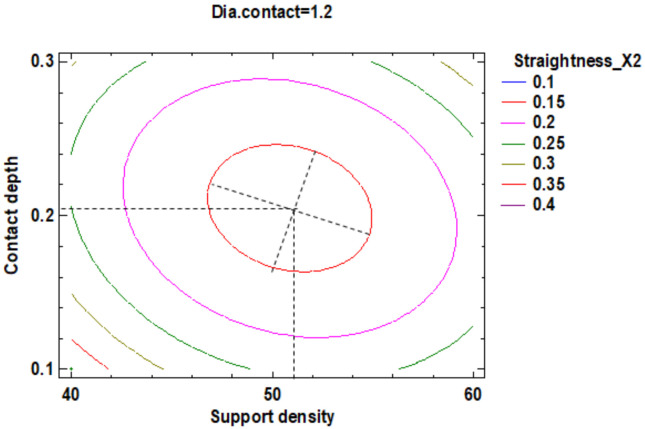
Contours of Estimated Response Surface for Straightness_X2 and centre diameter contact.

**Figure 19 polymers-12-01941-f019:**
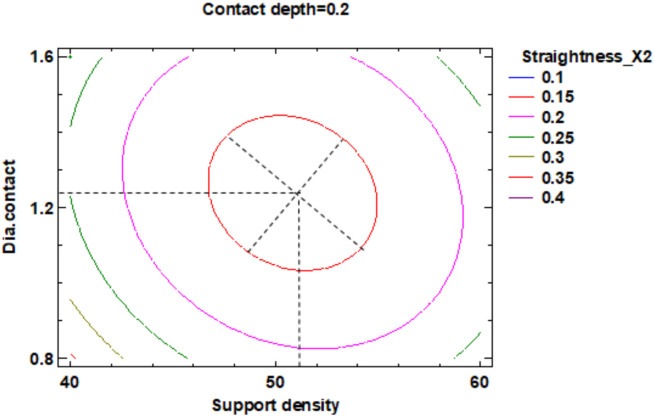
Contours of Estimated Response Surface for Straightness_X2 and centre contact depth.

**Figure 20 polymers-12-01941-f020:**
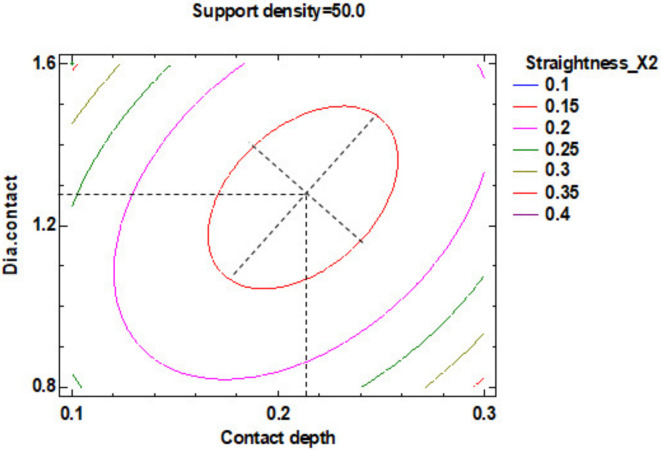
Contours of Estimated Response Surface for Straightness_X2 and centre support density.

**Figure 21 polymers-12-01941-f021:**
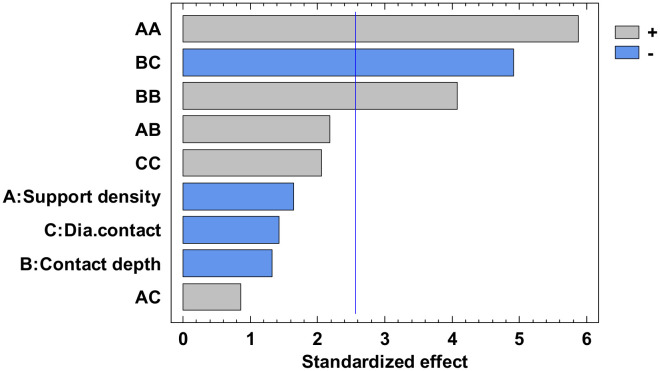
Standardized Pareto Chart for Straigtness_Y1.

**Figure 22 polymers-12-01941-f022:**
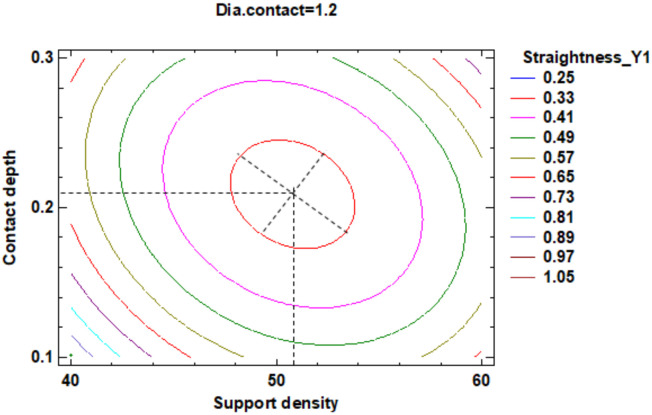
Contours of Estimated Response Surface for Straightness_Y1 and centre diameter contact.

**Figure 23 polymers-12-01941-f023:**
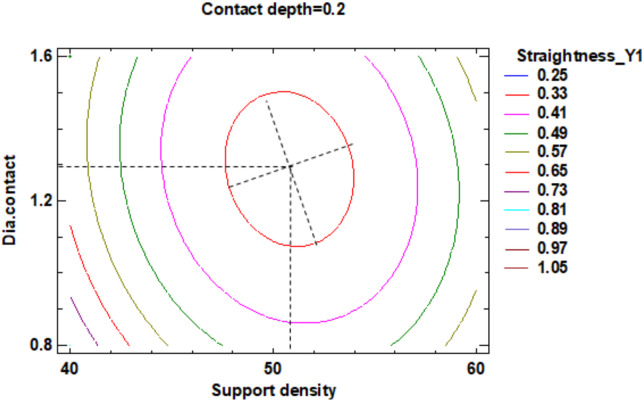
Contours of Estimated Response Surface for Straightness_Y1 and centre contact depth.

**Figure 24 polymers-12-01941-f024:**
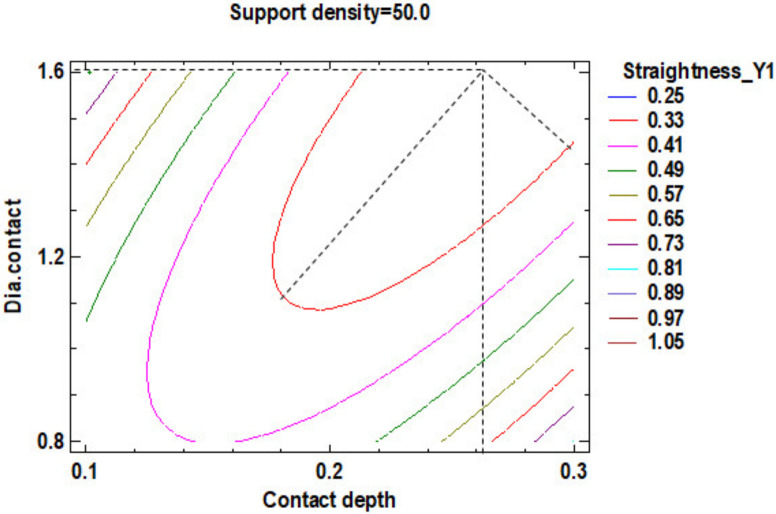
Contours of Estimated Response Surface for Straightness_Y1 and centre support density.

**Figure 25 polymers-12-01941-f025:**
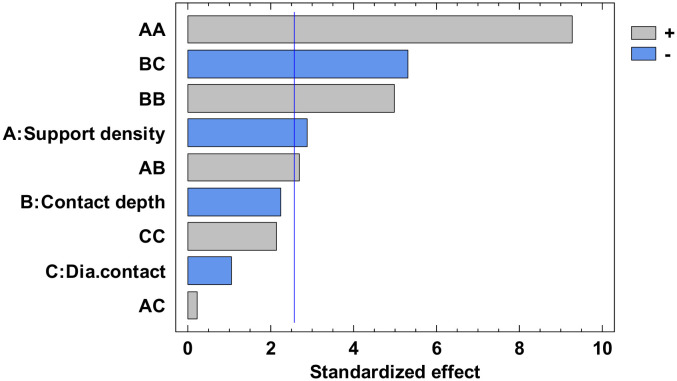
Standardized Pareto Chart for Straigtness_Y2.

**Figure 26 polymers-12-01941-f026:**
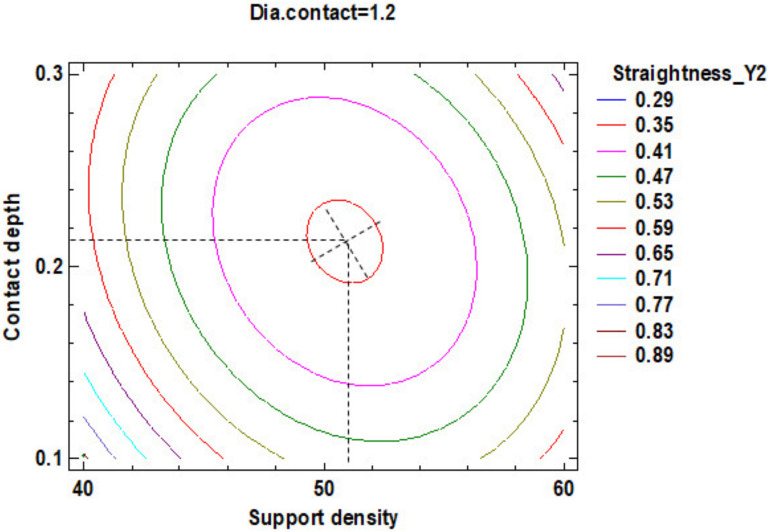
Contours of Estimated Response Surface for Straightness_Y2 and centre diameter contact.

**Figure 27 polymers-12-01941-f027:**
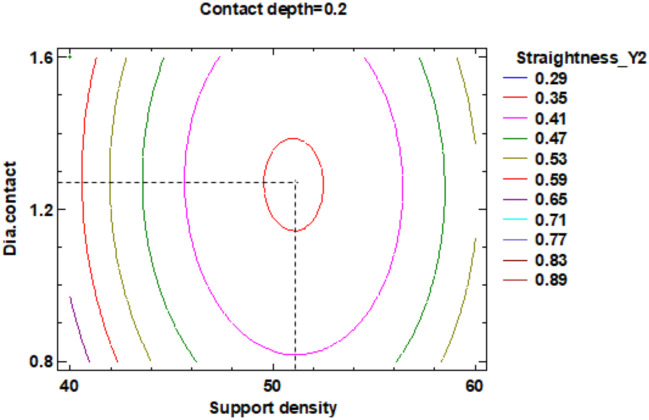
Contours of Estimated Response Surface for Straightness_Y2 and centre contact depth.

**Figure 28 polymers-12-01941-f028:**
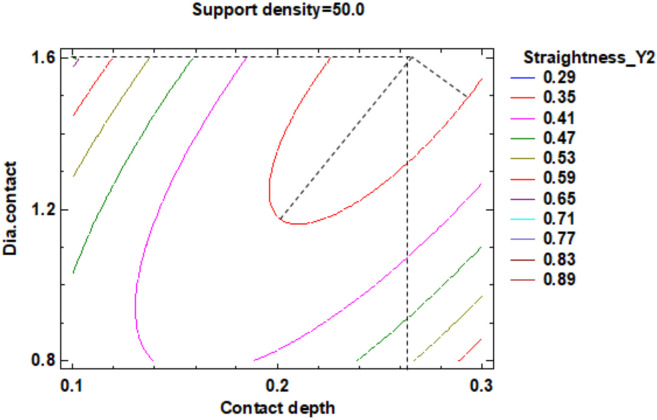
Contours of Estimated Response Surface for Straightness_Y2 and centre support density.

**Figure 29 polymers-12-01941-f029:**
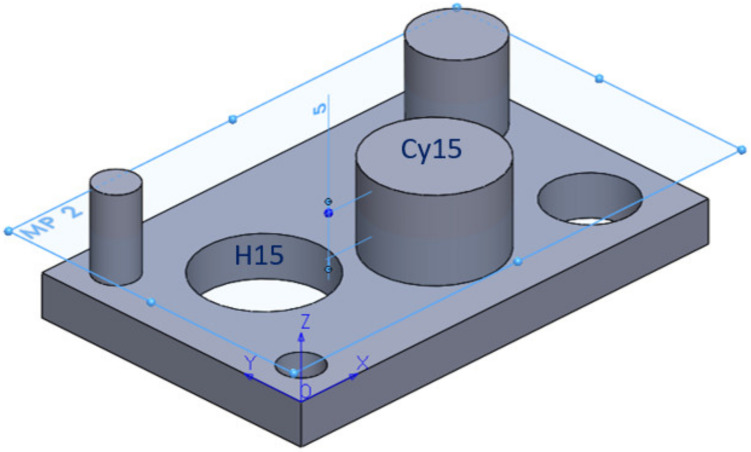
Definition of the cylindrical features and measuring plane.

**Figure 30 polymers-12-01941-f030:**
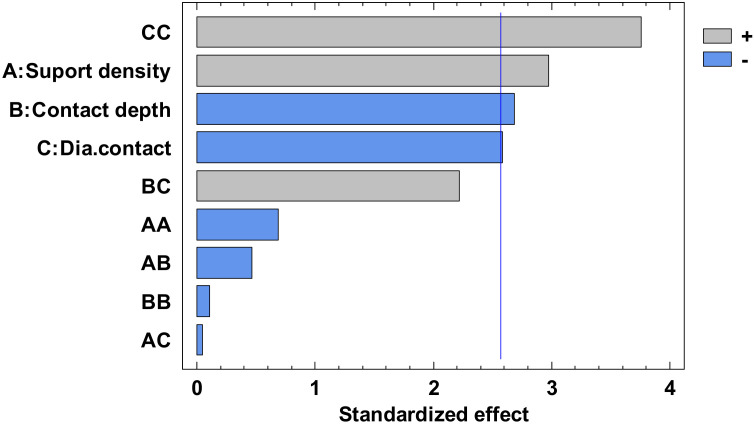
Standardized Pareto Chart for Roundness Hole_15.

**Figure 31 polymers-12-01941-f031:**
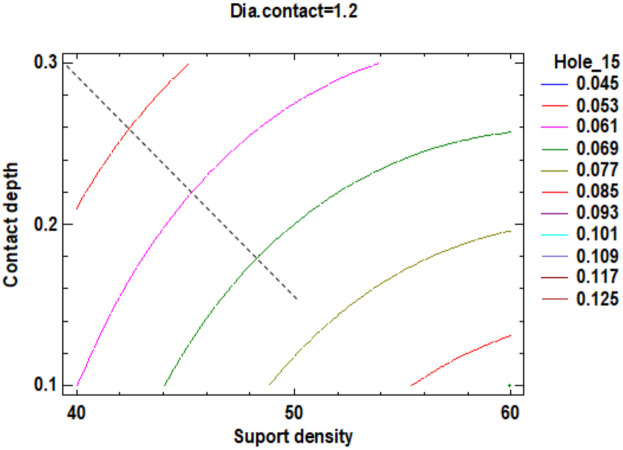
Contours of Estimated Response Surface for Roundness Hole_15 and centre diameter contact.

**Figure 32 polymers-12-01941-f032:**
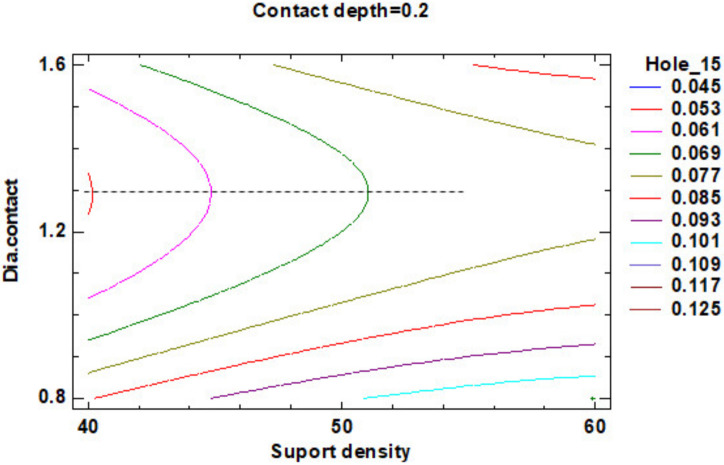
Contours of Estimated Response Surface for Roundness Hole_15 and centre contact depth.

**Figure 33 polymers-12-01941-f033:**
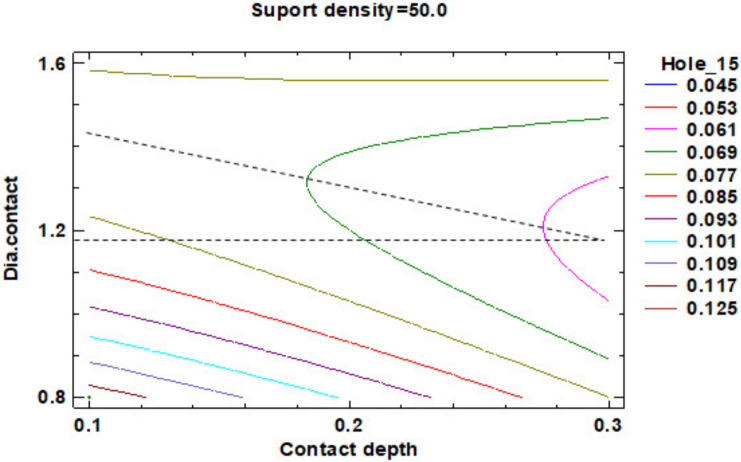
Contours of Estimated Response Surface for Roundness Hole_15 and centre support density.

**Figure 34 polymers-12-01941-f034:**
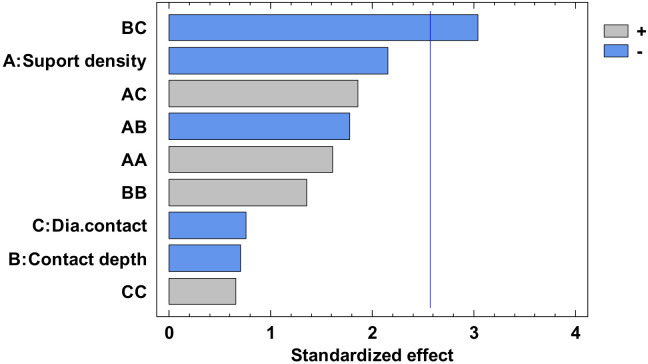
Standardized Pareto Chart for Roundness Circle_15_2.

**Figure 35 polymers-12-01941-f035:**
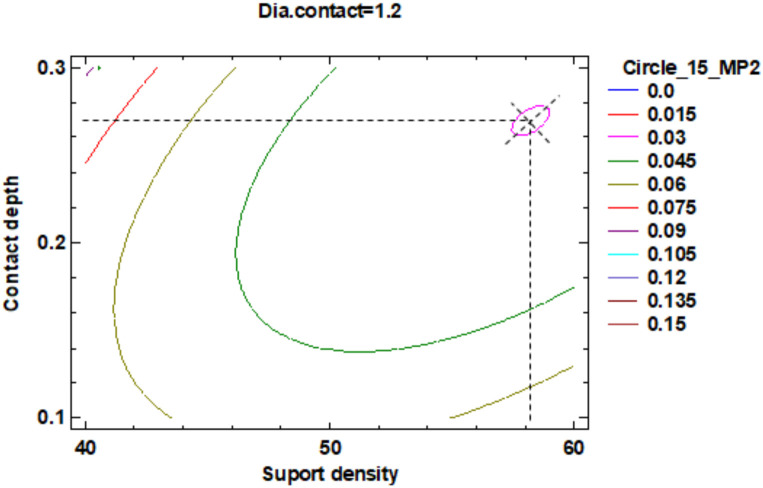
Contours of Estimated Response Surface for Roundness Circle_15_2 and centre diameter contact.

**Figure 36 polymers-12-01941-f036:**
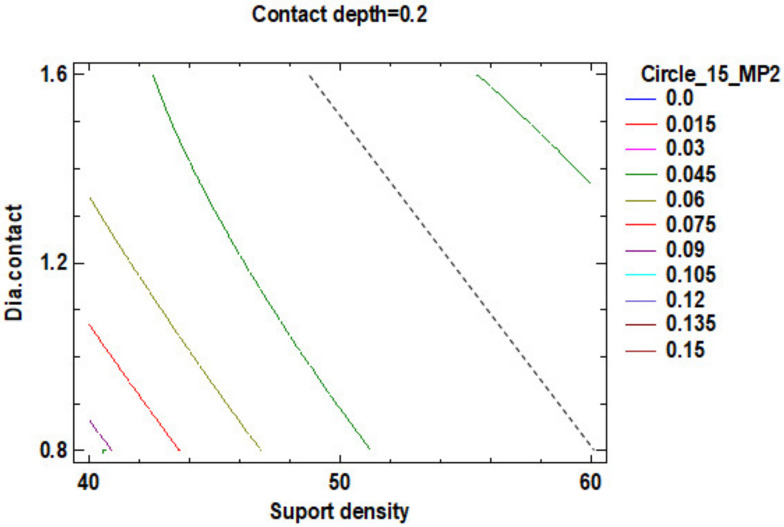
Contours of Estimated Response Surface for Roundness Circle_15_2 and centre contact depth.

**Figure 37 polymers-12-01941-f037:**
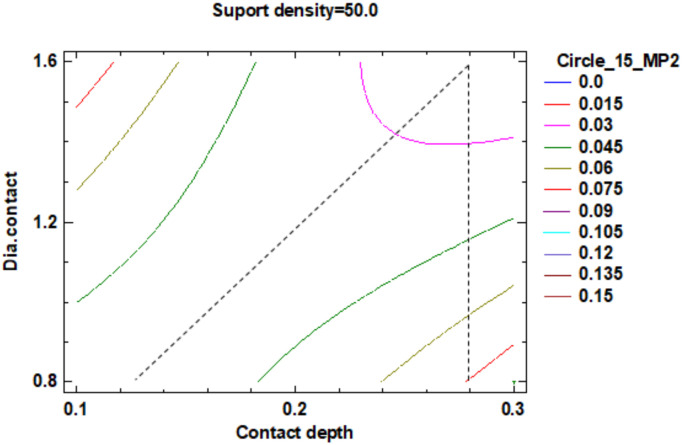
Contours of Estimated Response Surface for Roundness Circle_15_2 and centre support density.

**Table 1 polymers-12-01941-t001:** Analysis of Variance for Broken Supports.

Source	Sum of Squares	Df ^2^	Mean Square	F-Ratio	*p*-Value
A ^1^	14,210.7	1	14,210.7	76.53	0.0031
B ^1^	2646.0	1	2646.0	14.25	0.0326
AA ^1^	3872.0	1	3872.0	20.85	0.0197
AB ^1^	2862.25	1	2862.25	15.41	0.0294
BB ^1^	32.0	1	32.0	0.17	0.7060
Total error	557.083	3	185.694		
Total (corr.)	24,180.0	8			

^1^ A: Diameter contact, B: Support density, AA: Square Diameter contact, AB: Interaction Diameter contact with Support density, BB: Square Support density, ^2^ Df: Degrees of freedom.

**Table 2 polymers-12-01941-t002:** Analysis of Variance for Flatness.

Source	Sum of Squares	Df ^2^	Mean Square	F-Ratio	*p*-Value
A ^1^	0.0770667	1	0.0770667	18.86	0.0225
B ^1^	0.0039015	1	0.0039015	0.96	0.4005
AA ^1^	0.00642222	1	0.00642222	1.57	0.2987
AB ^1^	0.00245025	1	0.00245025	0.60	0.4951
BB ^1^	0.0200667	1	0.0200667	4.91	0.1134
Total error	0.0122555	3	0.00408518		
Total (corr.)	0.122163	8			

^1^ A: Diameter contact, B: Support density, AA: Square Diameter contact, AB: Interaction Diameter contact with Support density, BB: Square Support density, ^2^ Df: Degrees of freedom.

**Table 3 polymers-12-01941-t003:** Value of Experimental Factor for Flatness.

Experimental Factor Fix Factor	A ^1^ (mm)	B ^1^ (%)	C ^1^ (mm)	Source	Flatness (mm)
A ^1^	1.20	51.50	0.22	Figure 9 and Appendix A	0.4
B ^1^	1.30	51.50	0.20	Figure 10 and Appendix A	0.4
C ^1^	1.55	50.00	0.26	Figure 11 and Appendix A	0.4

^1^ A: Diameter contact, B: Support density, C: Contact depth.

**Table 4 polymers-12-01941-t004:** Value of Experimental Factor for Straightness X1.

Experimental Factor Fix Factor	A ^1^ (mm)	B ^1^ (%)	C ^1^ (mm)	Source	Straightness (mm)
A ^1^	1.20	51.00	0.21	Figure 14 and Appendix A	0.13
B ^1^	1.20	51.00	0.20	Figure 15 and Appendix A	0.13
C ^1^	1.28	50.00	0.21	Figure 16 and Appendix A	0.13

^1^ A: Diameter contact, B: Support density, C: Contact depth.

**Table 5 polymers-12-01941-t005:** Value of Experimental Factor for Straightness X2.

Experimental Factor Fix Factor	A ^1^ (mm)	B ^1^ (%)	C ^1^ (mm)	Source	Straightness (mm)
A ^1^	1.20	51.00	0.20	Figure 18 and Appendix A	0.1
B ^1^	1.20	51.00	0.20	Figure 19 and Appendix A	0.1
C ^1^	1.25	50.00	0.21	Figure 20 and Appendix A	0.1

^1^ A: Diameter contact, B: Support density, C: Contact depth.

**Table 6 polymers-12-01941-t006:** Value of Experimental Factor for Straightness Y1.

Experimental Factor Fix Factor	A ^1^ (mm)	B ^1^ (%)	C ^1^ (mm)	Source	Straightness (mm)
A ^1^	1.20	51.00	0.21	Figure 22 and Appendix A	0.25
B ^1^	1.30	51.00	0.20	Figure 23 and Appendix A	0.25
C ^1^	1.60	50.00	0.26	Figure 24 and Appendix A	0.25

^1^ A: Diameter contact, B: Support density, C: Contact depth.

**Table 7 polymers-12-01941-t007:** Value of Experimental Factor for Straightness Y2.

Experimental Factor Fix Factor	A ^1^ (mm)	B ^1^ (%)	C ^1^ (mm)	Source	Straightness (mm)
A ^1^	1.20	51.00	0.21	Figure 26 and Appendix A	0.29
B ^1^	1.35	51.00	0.20	Figure 27 and Appendix A	0.29
C ^1^	1.60	50.00	0.26	Figure 28 and Appendix A	0.29

^1^ A: Diameter contact, B: Support density, C: Contact depth.

**Table 8 polymers-12-01941-t008:** Value of Experimental Factor for Hole 15 mm.

Experimental Factor Fix Factor	A ^1^ (mm)	B ^1^ (%)	C ^1^ (mm)	Source	Roundness (mm)
A^1^	1.20	40.00	0.30	Figure 31 and Appendix A	0.045
B^1^	1.30	40.00	0.20	Figure 32 and Appendix A	0.045
C^1^	1.20	50.00	0.30	Figure 33 and Appendix A	0.045

^1^ A: Diameter contact, B: Support density, C: Contact depth.

**Table 9 polymers-12-01941-t009:** Value of Experimental Factor for Cylinder 15 mm.

Experimental Factor Fix Factor	A ^1^ mm	B ^1^ (%)	C ^1^ (mm)	Source	Roundness (mm)
A^1^	1.20	58.00	0.27	Figure 35 and Appendix A	0.015
B^1^	0.80	60.00	0.20	Figure 36 and Appendix A	0.015
C ^1^	1.60	50.00	0.28	Figure 37 and Appendix A	0.015

^1^ A: Diameter contact, B: Support density, C: Contact depth.

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
