# Peer review of "Study of the Influence of Technological Parameters on Generating Flat Part with Cylindrical Features in 3D Printing with Resin Cured by Optical Processing"

_polymers, 2020, doi:10.3390/polym12091941_

Round 1
Reviewer 1 Report
See attached

Author Response
We would like to thank you for your review.

Reviewer 2 Report
The research on the influence of DLP 3D printing process on the characteristics of the flat and cylindrical surfaces can provide help for DLP 3D printing and other 3D printing methods that need to add supporting structure. It is suggested that the research be received after minor revise. The following are some suggestions for the author.
1.There is a lack of comparison with similar studies in the past.
2.The contact depth in lines 338 and 360 are expressed incorrectly.
3.The article should add some more formulas. For example, the origin of R2 is not clear.
4.In Figure 6, the meanings of A and B should be indicated separately, and the meanings of AA and AB should be supplemented.
5. The title angle in the X and Y axes in Figure 8 is not consistent with the coordinate axis.
6.In Figure 9, the number and axis overlap.
7. The size, font and spacing of pictures are not uniform.
8. There are redundant parts in the picture,such as figure 19 and figure 21.
9.There is a problem with typesetting in the discussion part of the article.
Author Response
We appreciate your valuable comments.
We would like to thank you very much for your review.
Manuscript ID: polymers-892066
Type of manuscript: Article
Title: Study of the Influence of Technological Parameters on Generating Flat Part with Cylindrical Features in 3D Printing with Resin Cured by Optical Processing
Authors: Aurel Tulcan *, Mircea Dorin Vasilescu *, Liliana Tulcan *
Received: 22 July 2020, Response from reviewer: 04 Aug 2020 10:42:47, Received: 07 Aug 2020
Response to reviewer: 14.08.2020 (GMT +2) 17:00,
Response to Reviewer 2 Comments
Point 1: There is a lack of comparison with similar studies in the past. 

Response 1: For the DLP 3D printing, from the technological and structure of supports point of view, there are no studies carried out in the past, that be known by the authors of this paper.
However, we modified the first part of the Article and we added, from 1 to 10 and 13 to 17 references that are close to the research area of this paper.
Point 2: The contact depth in lines 338 and 360 are expressed incorrectly.
Response 2: Sorry for this error. We corrected.
However, for better visibility of the experimental research data, the results from this part of the research were centralized in a few tables. For example, these modifications can be seen in lines 369 or 385 of modified paper.
Point 3: The article should add some more formulas. For example, the origin of R2 is not clear.
Response 3: As one of the evaluators invoked the large size of the article, the part referring to R2 was removed from the text. However, the scientific result of the research was not reduced or affected, because we have put an accent on the aspects referring to the P-Value and Pareto Chart, resulting in conducting the study with the ANOVA part of mathematical analysis.
Point 4: In Figure 6, the meanings of A and B should be indicated separately, and the meanings of AA and AB should be supplemented.
Response 4: Because the data processing program does not allow changes on the graphical side, the mentioned aspect was corrected by adding in the tables footer the meanings of the factors (A, B and C) and their interactions (AA, AB, ...) that solves in our opinion this aspect. As example, these modifications can be seen in lines 369 or 385 of modified paper.
Point 5: The title angle in the X and Y axes in Figure 8 is not consistent with the coordinate axis.
Response 5: As one of the evaluators invoked the large size of the article and number of Figures, the part referring to Response Surface was put in the Appendix A. It should be noted that these graphs were used to better understand how to generate minimum point to determine the coordinates of points in the specific graphs Contours of Estimated Response Surface. The following graphs were completed with dash lines for determining the point values and the guidelines for identifying the point in the graphs.
Point 6: In Figure 9, the number and axis overlap.
Response 6: The mentioned problem was corrected in all the graphs in which this double notation appeared by canceling the values inside the graphs and keeping the ones from the legend. All graphs named Contours of Estimated Response Surface were completed with the dash lines for determining the point values and the guidelines for identifying the point in the graphs.
Point 7: The size, font and spacing of pictures are not uniform.
Response 7: The mentioned aspects were remedied and in the same time, the whole paper was rechecked from all the points of view of the settings including images, equations and tables.
Point 8: There are redundant parts in the picture,such as figure 19 and figure 21.
Response 8: The mentioned observation was corrected by deleting from the text describing the figure from all the places where this observation occurs.
Point 9: There is a problem with typesetting in the discussion part of the article.
Response 9: The problems are in the Discussion 4.3 and it is by Tab and dimension of later. These problems were corrected.
Due to the momentary impossibilities of loading the paper on the page, I attached to the message the new version of the paper. With thanks.

Reviewer 3 Report
Dear authors
While the experimental method used is simple, it is easily understandable in it approach to deformation measurements. It is thus helpful in identifying and better understanding the impact supports have on the various features of a SLA-built part.
I have however the following remarks to improve the quality of the paper:
- You have mentioned the meshing refinement in section 2.3 and developed a bit on that. There is however no follow-up in the experimental section on any possible effects. Experimental data are required to be added, or this topic could be withdrawn from this paper and expanded in further studies.
- There are too many figures in section 3.2, making it hard for the reader to follow through your line of thoughts. While those figures are helpful in identifying trends, some of these could be placed as Supplementary material, keeping the meaningful figures in the text.
- There is no conclusion per se, only the Discussion section. A general conclusion needs to be added to summarize the work. Only tidbits of conclusion are found in the various parts of section 4.
- The comments found in the discussion section are sometime hard to understand and visualize on the piece, especially when a specific position is mentioned. Maybe a figure with the piece and some identifiers would be helpful.
- Some of the comments in the discussion need to be rewritten as they are more descriptive than trying to explain a phenomenon.
- Proof-reading by a native or fluent English speaker is needed to correct the text.
Author Response

(The authors gave the same response as above.)
